# Post-Implementation ERP Software Development: Upgrade or Reimplementation

Adam Domagała *, Katarzyna Grobler-Dębska ⬤, Jarosław Wąs ⬤ and Edyta Kucharska ⬤

Faculty of Electrical Engineering, Automatics, Computer Science and Biomedical Engineering,
AGH University of Science and Technology, 30-059 Krakow, Poland; grobler@agh.edu.pl (K.G.-D.);
jarek@agh.edu.pl (J.W.); edyta@agh.edu.pl (E.K.)
* Correspondence: adomagala@agh.edu.pl

**Abstract:** The paper deals with problems in the post-implementation phase of management Enterprise Resource Planning (ERP) systems. Proper management of the system maintenance stage is a basis for efficient system development in terms of business needs. Based on the research and analysis of collected materials, it turns out that making a decision to upgrade the system is equally crucial. We present revealed mechanisms determining the post-implementation approach to upgrade or reimplement the ERP system. The main aim is to determine the methodology and difference understanding to achieve success in the post-implementation stage. The paper shows that the systemic approach to the maintenance stage of the ERP system affects its further decisions: upgrade or reimplement. It has a direct impact on future maintenance costs and the scope of new business demands. This research is an outcome of industry–academia collaboration and based on several developed implementation systems, achieved upgrade and reimplementation projects. Based on case study analysis, we show that reimplementation means an evolution of the current ERP processes rather than another attempt to "reimplement" an unsuccessful system implementation. On the other hand, upgrades are not only a tool or system actualization but the easiest way to bolster company sustainability and to have the information system up to date. The issues discussed in the article will be used to develop changes in the implementation methodology of ERP systems.

**Keywords:** Enterprise Resource Planning (ERP); software upgrade; software reimplementation; post-implementation methodology; management and empirical research methods

## 1. Introduction

Since the 1990s, implementation of Enterprise Resource Planning (ERP) systems have been a major method to improve competitiveness, optimize, and boost operational performance [1]. Therefore, many, large companies have implemented ERP systems [2,3]. On the other hand, the ERP penetration in small and medium-sized enterprises (SMEs) also increased steadily [4]. Companies fulfil this gap with final closure and consider it as the end of activity rather than a life cycle phase [5]. ERP systems are currently present in at least 75% of North American manufacturing firms [6] and in Top Fortune 500 businesses and international organizations [2]. In order to manage continuous benefits of an ERP implementation, several aspects of the system evolution must be taken under consideration [7]. This evolution consists of multiple iterations such as revisions, updates, upgrades [8]. Chief Information Officers(CIOs) focus on keeping the ERP system up to date and are ready for future organisation business needs.

Organized actions connected with the implemented ERP system evaluation are sus-tainability initiatives. In the paper [9], authors present post-implement amendment (ERP-PIA) frameworks and identify three categories of amendment activities: maintenance, upgrade, enhancement. These activities can improve performance and build a trustwor-thy reputation for company contractors. In the long run, the lack of revisions, updates, and upgrades leads to system reimplementation. Such an operation is extravagant in

terms of time, cost, and human resources. ERP reimplementation projects are the most serious causes of wastage in continuous improvement and sustainability. We argue the ERP reimplementation differs from maintenance, upgrade or enhancement of ERP-PIA categories and is caused by neglect in the post-implementation phase. Our motivations are creating CIO's awareness of activities in the post-implementation stage and defining a crucial moment for a technical upgrade. Additionally, we would like to indicate proper managing importance in a company's crucial system and underline the understanding of the necessity of maintenance activity.

The main contributions of this study are:

- Identification of risks resulting from negligence during the maintenance phase;
- Proposal classification: extensions, patches, updates, upgrades and reimplementation;
- Convergence identification between no continuous ERP improvement and a necessity to reimplement the system;
- Comparative analysis of methodologies according to post-implementation phase.

An integrator supplier experience shows that a significant part of companies, after successful implementation, have not shown commitment to further development of the system. The research shows the lack of software engineering in companies to impacts on their technological dept and grow. The presented analysis is the basic research to develop a methodology for conducting post-implementation projects based on empirical research. The post-implementation methodology will be used in new industry projects to increase knowledge dissemination and minimize software development gaps. The research reveals that the companies which have not upgraded their ERP software for at least three released next versions of the system have lost the ability to develop their software and the reimplementation project is obligate.

The article is organized as follows: in the section "Implementation and maintenance of the ERP system background" we present an overview of the literature on actual methodologies for ERP implementation projects and common types of ERP maintenance activities. Presented methodologies are proposed in scientific articles or by the top ERP software vendors and they focus on new implantation projects, depreciating the importance of the management post-implementation phase. In the section "Methodology and data collection" we describe how we conducted the research. We analyzed ERP post-implementation projects in which the first author was the project manager. We selected evaluation projects in which previous successful implementation has been carried out. Based on these analyses, we also propose various types of post-implementation projects and phases of these projects. The section "Analysis and results" includes a comparison of the proposed types of post-implementation projects and their importance for the company's sustainable development. The section "Discussion" includes the findings and the last section contains conclusions and future work.

## 2. Implementation and Post-implementation ERP System Background

"ERP are commercial software packages that enable the integration of transaction-oriented data and business processes throughout an organization" [10]. In the literature, we can find other definitions of the ERP system [6,11]. Mainly, ERP is software designed to support managing organization processes across most areas of their businesses. Depending on the vendor, system functionalities consists a list of typical components such as: accounting, distribution, finance, human resources, logistics, manufacturing, production, purchase, and sales [12]. For decades, vendors have been developing fundamental packages with at least every 3 years a major upgrade and several small updates/legal patches to keep the system up to date and running smoothly [13]. It provides functionality evolution to customize user interfaces and no-code tools for creating fields, user screens, menu and reports. Major upgrades often reveal new module functionalities or business processes. Due to the significant market saturation, suppliers are constantly developing software and adding modules that try to respond to client expectations. It is essential for small and medium-sized enterprises(SME) as well as for global organizations to have

an up-to-date ERP system. It is necessary for SME to deal with tasks in an efficient and practical way to operate competitively in an international and local market, which can be achieved by systematic task analyses in relation to company organizational and economical abilities. Implementation assessment depends on a company intention to use the ERP system to achieve strategic goals [14]. The paper examines long-term effects of the ERP system revision [15] using the Balanced Scorecard method (BSC) for SMEs. In the real case study, the ERP system has impacted enhanced revenue opportunities and improved market positioning. SMEs, unlike large companies, do not have capital to purchase such tools and a full implementation. Moreover, large enterprises have greater opportunities to use new technologies and transform them into smart factories [16]. The post-implementation phase should improve defined indicators in each BSC perspective. The analysis shows that in a given company there is a need to develop ERP system functionality. Moreover, ERP systems are a backbone for Factory of the Further (FoF) [17]. FoF is based on a solid foundation regarding to modularization, mass customization, distributed control, IoT, modelling and forecasting, collaboration, distributed SC, distributed manufacturing, and transparent processes.

Recent years show ERP implementations are moving into the cloud because of client needs. It consists of purchasing/having access to selected services and ERP system functionalities. One of the main reasons that this approach is an easier route for maintenance, being up to date with functionality and potentially ready to extension ERP with occurring business needs. This is the main advantage in comparison with on-premise solution and ERP installed on a private cloud. On-premise solution is often chosen for significant ERP system customization and integration with other systems in the company. These types of implementation projects involve high risk. In [18] authors presents a review of journal publications on ERP and Information Systems (IS) integration. Authors show that in the post-implementation stage the same systems integration is needed. A cross-analysis between the implementation stage and integration type was presented in this paper. Moreover, the integration type, implementation project/stage in most research was found to be mainly considered. Only a few papers have studied all stages of the same research or maintenance phase. This confirms our belief that the post-implementation phase is often neglected.

### 2.1. ERP Implementation Projects

ERP system implementation is a unique [19] and formidable challenge, with a typical timing from one to five years [20]. Such projects are extremely risky, expensive and involve a lot of company resources. Literature analysis reveals that at least a quarter of the projects are failures [21]. This is only aware of the need to standardize the appropriate approach to the stages of implementation and maintenance of the ERP system.

The typical ERP implementation project is organized sequentially with different steps in available frameworks and methodologies. Table 1 lists implementation methodologies defined in scientific research. [22].

**Table 1.** ERP literature implementation models often omit the post-implementation activity phase. The table is based on [23].

| Author(s) | ERP Implementation Model |
|---|---|
| Bancroft et al. (1998) [24] | (1) Focus, <br> (2) Creating As-Is picture, <br> (3) Creating of the To-Be design, <br> (4) Construction and testing, <br> (5) Actual Implementation |

**Table 1.** *Cont.*

| Author(s) | ERP Implementation Model |
| --- | --- |
| Kuruppuarachchi (2000) [25] | (1) Initiation,<br>(2) Requirement definition,<br>(3) Acquisition/development,<br>(4) Implementation,<br>(5) Termination |
| **Markus and Tanis (2000)** [10] | (1) Project chartering,<br>(2) The project,<br>(3) Shakedown,<br>(4) **Onward and upward** |
| **Makipaa (2003)** [26] | (1) Initiative,<br>(2) Evaluation,<br>(3) Selection,<br>(4) Modification, Business process Re-engineering, and Conversion of Data,<br>(5) Training,<br>(6) Go-Live,<br>(7) Termination,<br>(8) **Exploitation and Development** |
| **Parr and Shanks (2000)** [27] | (1) Planning,<br>(2) Project: setup, re-engineer, design, configuration and testing, installation,<br>(3) **Enhancement** |
| **Ross (2000)** [28] | (1) Design,<br>(2) Implementation,<br>(3) Stabilization,<br>(4) **Continues improvement**<br>(5) **Transformation** |
| **Umble et al (2003)** [29] | (1) Review the pre-implementation process to date,<br>(2) Install and test any new hardware,<br>(3) Install the software and perform the computer room pilot,<br>(4) Attend system training,<br>(5) Train on the conference room pilot,<br>(6) Established security and necessary permissions,<br>(7) Ensure that all data bridges are sufficiently robust and the data are sufficiently accurate,<br>(8) Document policies and procedures,<br>(9) Bring the entire organization online, either in a total cut over or in a phased approach,<br>(10) Celebrate,<br>(11) **Improve continually** |
| Verviell and Halingten | (1) Planning,<br>(2) Information search,<br>(3) Selection,<br>(4) Evaluations,<br>(5) Negotiation |

Due to the subject discussed in this paper, it should be underlined that not all methodologies define the stage of post-implementation activities. In the analyzed literature, the post-implementation stage has been identified by [27] as "Enhancement" [29], as "Improve continually" [26], as "Exploitation and Development" [10], as "Onward and Upward" and [28] as "Continues Improvement and Transformation". These methodologies are highlighted in Table 1 and show a raise awareness of continuous improvement activities. In the newest research papers [9,30–32], the post-implementation activities are of interest.

Despite the scientific framework and methodologies, almost all of ERP leader vendors have their own methodology. Their foundations lie in the scientific knowledge, but in

detail they are being customized for their purpose. The great number of integrations which have to be done with ERP projects during implementation also play an important role [33].

As mentioned in Table 2 the methodologies are sequential and, on many levels, based on generally applicable standards in project management, such as PMBOOK or PRINCE2.

**Table 2.** Top ERP vendors' implementation methodologies.

| Vendor Name | Methodology |
| --- | --- |
| IFS | IFS Implementation Methodology, with these phases: Initiate Project, Confirm Prototype, Establish Solution, Implement Solution, Go Live. |
| INFOR | Infor Deployment Method, with five phases of the implementation process: Inception, Elaboration, Construction, Transition and Optimization. |
| MICROSOFT | MS Dynamics Sure Step Methodology, with these phases: Diagnostics, Analysis, Design, Development, Deployment, Operation. |
| ORACLE | A.I.M. (Applications Implementation Methodology), with these phases: Definition, Operations Analysis, Solution Design, Building, Transition, Production. |
| SAP | ASAP methodology, with these phases: Project Preparation, Blueprint, Realization, Final Preparation, Go Live Support, Operation. |

Customized internal ERP vendors' methodologies are up to date with an experience after successful as well as failed system implementations. Most popular vendors' methodology is based on waterfall methodology with different steps in their phases. Integrators of ERP systems also adopt agility into their methodologies. It is most popular with large projects with an enormous number of customizations/modifications. That is why the phase named "development", "modification", "business processes reengineering", "conversion of data" and "preparing of the solution" often consists of a sequence of popular iterations such as "sprint", "wave", "iteration".

*2.2. ERP Post-Implementation Activities*

Over the last two decades, the primary IT system market supporting management in large and medium-sized organizations has been saturated. Currently, the main activity is to reimplement or upgrade the previously implemented system. However, most implementation methodologies focus on a phases approach to implement the ERP system. Companies (SME and large) are faced with ERP maintenance and development needs. The end of the implementation process opens the post-implementation phase. It is considered that IT may have a critical role in shaping ERP post-implementation adaption. In the paper [32], authors present control variables which include: IT turbulence, IT mindfulness, IT financial, adaptive agility, firm size and time since ERP implementation. It is assumed that the longer ERP has been deployed, the more likely a company adapts the ERP to their business needs. The paper analyzes action impact over time and its impact on the future decisions with ERP adoption.

From the first days of the starting ERP system, CIOs mainly pay attention to maintain implemented solutions by receiving patches (fixes) for identified bugs or legal changes which impact the implemented ERP functionalities. Patches are easy to install and tests are essentially focused on the process part because it is usually built by providers on demand or forced by a legislator. Every few months, ERP companies provide updates for their actual ERP system. It consists of new functionalities, legal changes, and bug fixes which appeared after the last released package. Update installation is similar to the patches installation process, but it takes more time and demands complex tests across all implemented modules and ERP processes before launching into the production environment. Depending on the ERP provider, every few years they release a new version of a system. The upgrade installation process is often managed by the system provider or integrator. The upgrade process takes a few months; it mostly depends on ERP system and implementation procedures in

companies. Upgrade is still only a technical process and all data are being automatically moved from an old to a new version. All activities are connected to on-premise installations on a client infrastructure or private cloud.

### 2.2.1. ERP Upgrades

Upgrades are defined in technical or functional terms and are vendor-provided packages. The scope of a technical upgrade helps to keep the system in a supported version, while functional upgrades refer to software functionality that can lead to business process improvement [9].

Vendors describe technical upgrades as a technical activity, which moves the currently implemented system to an updated version and technology ERP platform. Technical upgrades are important when the current system version is going to expire. The demand of businesses is to have legal and technical support, while the IT department wants to keep abreast of advancing technology [34].

The reason to proceed with functional upgrades is to extend business functionality and they are often initiated by the "business" unlike technical updates which are initiated by the ERP support (IT) department.

Regardless of the above approach, "implementing a completely new system (even if it is similar) is considered outside the scope" of an upgrade [9].

### 2.2.2. ERP Re-Implementation—Enhancement Functional Upgrades

Functional upgrades are usually preceded by technical upgrades and are expected to have high technical improvement and business change [35]. In ERP system, enhancements are considered with add-ons and upgrades [36] that provide additional business functionality.

The upgrade approach involves merging customizations and transferring data between ERP versions. Reimplementation involves installing the latest ERP version and requires setting up of the system and databases as a new software. The reimplementation project can be defined as midway between "complete replacement of a legacy system" and "technical upgrade" [14]. It implies a major re-engineering effort in business processes and data conversion [37].

## 3. Methodology and Data Collection

We conducted the research based on five companies using the information collected during reimplementation or upgrade projects. Data were collected through observation of steering committees, periodic progress reports, support system data/issue logs, study of formal documentation (meeting reports, protocols, list of change request), and interviews with client project managers, system integrator ERP consultants and CIOs. Due to diverse project scopes, the conducted research could not be carried out based on a standardized questionnaire/survey. Information was gathered based on vendor document templates during post-implementation projects managed by the first author.

In following analyzed projects, the schedule and scope of all activities in a post-implementation were organized with CIO strategies. Implementation, reimplementation, or installation procedures are provided by ERP vendor companies and different approaches were used in the analyzed companies. Collected data and information were gathered by the authors during several implementations, upgrades, updates, reimplementations and re-engineering activities with all clients. Three of the five companies are manufacturing and two of them are from the utilities sector.

All surveyed companies are large Polish enterprises that use the same ERP system. This system is dedicated to large and medium-sized enterprises and is one of the top 10 most popular ERP systems on the global market [38]. Information about organizations are provided below:

**Company 1:** A company from the utilities industry that has been operating in the current form for over 70 years. The company employs around 800 employees. The

ERP system is implemented in the following areas: finance and accounting, human resources, maintenance, sales, purchasing, and payroll.

**Company 2:** A company from the utilities industry that has been operating for over 80 years on the Polish market. The company employs around 400 employees. The ERP system is implemented in the following areas: finance and accounting, human resources, maintenance, sales and operations planning, payroll, purchasing, and warehouse. The ERP is integrated with GIS.

**Company 3:** The company was founded over 100 years ago as a producer of modern solutions in the field of sanitary and heating fittings. The company employs over 1,000 employees in three production plants in Poland. The ERP system is implemented in the following areas: finance and accounting, human resources, supply chain, sales and operations planning, manufacturing, production planning, and payroll. The ERP is integrated with EDI.

**Company 4:** The company was founded in 1990 as a family business. The company produces and exports solutions for aluminum systems to more than 50 countries. The company employs over 500 employees in 2 production plants in Poland. The ERP system is implemented in the following areas: finance and accounting, human resources, supply chain, sales and operations planning, manufacturing, production planning, and payroll. The ERP is integrated with EDI automated warehouse and sales tools.

**Company 5:** The company was founded as a family business and has been producing and selling furniture and office equipment for about 30 years. The company employs over 300 employees in 2 sites. The ERP system is implemented in the following areas: CRM, finance and accounting, human resources, supply chain, sales and operations planning, manufacturing, and production planning. The ERP is integrated with MES.

The selection was made from over several dozen projects implemented in the last 15 years. The selection was based on a comparison of projects/implementations that were carried out by one integrator implementation team. The purpose of this action was to eliminate the impact of competence and incompetence of the implementation team on the project quality. All considered projects were implemented based on the same implementation methodology. The study does not analyze implementation costs, reimplementation and upgrade. This is due to the fact that each of the companies implemented a different scope of the project, based on various rates in varied periods of time.

From the organization perspective, it is essential that the upgrade activity, regardless of the optimization scope, lasts up to 9 months. However, the activity time depends on the potential scope of changes and extensions as well as test procedures planned as part of the project. Modules and functionalities which were provided during all ERP implementation activities are listed in the Table 3. We used the following abbreviations of modules' names: FIN—finance; HR—human resources; SCM—supply chain management; S& S—sales and service; MFG—manufacturing; CRM—customer relationship management; B2B—business to business; WF—workflow, and the following abbreviations of functionalities' names: MRP—material requirements planning (functionality which is part of the MFG module) and systems which aren't part of ERP: WMS—warehouse management system; GIS— geographic information system.

**Table 3.** Scope of work—analyzed companies.

| Company | ERP Implementation | ERP Upgrade 1 | ERP Upgrade 2 | ERP Reimplementation |
|---|---|---|---|---|
| Company 1 | In 2008/2009, scope: FIN, HR, SCM, S & S modules | Upgraded in 2013 with a version of the application with a small optimization in processes | Upgraded in 2018 with version of the application with a small optimization in processes and new PAYROL module | |
| Company 2 | In 2008/2009, scope: FIN, HR, SCM, S & S modules | | | In 2017/2018 new version of previous implemented modules and news was applied: GIS integration functionalities, mobile solutions, preventive maintenance |
| Company 3 | In 2005/2006, scope: FIN, HR, SCM, S & S, MFG modules | Upgraded in 2012 with a version of the application with a small optimization in processes | Upgraded in 2017/2018 with a version of the application and new functionalities: CRM, MRP, preventive maintenance, B2B, workflow | |
| Company 4 | In 2002/2003, scope: FIN, HR, SCM, S & S, MFG modules | has been stopped | | In 2018/2019 new version of previous implemented modules with big changes in all processes and new functionalities such as PAYROL, CRM and integration with WMS was applied |
| Company 5 | In 2002/2003, scope: FIM, HR, SCM, S & S, MFG modules | has been stopped | | In 2019/2020, a new version of previous implemented modules with big changes in all processes and new B2B modules was applied |

Upgrades were conducted by companies at least every second new release, between three years (1 upgrade) to five–six years (2 upgrades) in two implementation projects. Every upgrade includes optimization and modifications to the core processes and functionalities. Companies decided to go through ERP reimplementation projects between 10 and 15 years after the successful first implementation project. In all three cases, there were any significant optimization activities in ERP implemented processes. In all these companies, managers decided to go through all processes again and fulfill actual expectations with a new version of the implemented ERP system.

In this chapter, we identify and measure main ERP activities. This identification is according to the literature and the best practices of the system support. Table 4 shows the classification of the type of work performed at the stage of system implementing and maintaining. The data were collected from interviews with business process managers (BPM), CIOs, and vendor technical consultants. The most important processes of the maintenance phase are highlighted. The first is "Infrastructure installation, company business processes analyse & implementation". The second activity related to the ERP system development is its optimization through changes at the level of the entire organization named

as "functional change optimization in module with influence on company processes". Further activity is related to the continuous improvement of the application at particular modules named as "new functionalities in a tight field period package with legal changes and bug fixes". Due to the importance of system maintenance, activities carried out on the business demand were separated, named "optimization—on request changes with impact on company business processes or local functionality". The remaining group of activities is related to the typical maintenance work: "legal changes, bug fixes (often on demand) patches".

**Table 4.** Analysis of post-implementation activities compared with SOW.

| Scope of Work (SOW) | ERP Extension (on Request) | Patches (Fixes) | Updates | Upgrades | ERP Reimplementation |
|---|---|---|---|---|---|
| Infrastructure installation, company business processes analyses and implementation | | | | | ● |
| Functional changes, optimization in module functionalities with influence on company processes | | | | ● | ● |
| New functionalities in a tight field, period package with legal changes and bug fixes | | | ● | ● | ● |
| Optimization—on request, changes with impact on company business processes or local functionalities | ● | | ● | ● | ● |
| Legal changes, bug fixes (often on demand) patches | ● | ● | ● | ● | ● |

All respondents explain that they planned patches and update installation activities in their day-to-day work routine. Only one of five companies was up-to date with the upgrade installation, and the rest of them concentrated on the legal changes and bug fixes installation. Optimization in the company with the current ERP system version was being done smoothly during upgrade periods. Due to the very long time between the first successful implementation and the need to update the system, which often lasted over 10 years, the ERP system required reimplementation. Some organizations were obliged by businesses to reimplement due to changes in the company environment and work scope. Most of ERP providers release updates for their current version of the system and the previous one. When the next application version appears, vendors discontinue support (update releases) for the third version in a row. Three sample clients postponed their upgrade activities and after the standard support period concentrated on bugs identified and eventually patch installations.

## 4. Analysis and Results

Implementation works involve organization resources, with the most important being human resources. Due to the complexity and comprehensive nature of the implementation, the most expensive and time-consuming are implementation and reimplementation works. Total execution time of individual work analysis related to the upgrade showed that they lasted longer than one reimplementation project. Activity duration is presented in Table 5; however, it should be taken into account that upgrade activities are continuous improvement and care for organization sustainability. In all projects, repeatable activities consumed a similar amount of time. Patches take approximately up to two weeks. Updates depend on the internal type and test procedures and took up to two months. Analysis

shows that the upgrade activities last from six to nine months. The first company upgrade took three months more than the second one in both cases (see details in Table 5).

**Table 5.** Clients' project activity duration in months.

| Activity Type: | ERP Implementation | ERP Upgrade 1 | ERP Upgrade 2 | ERP Reimplementation |
|---|---|---|---|---|
| Company 1 | 15 | 9 | 6 | |
| Company 2 | 18 | | | 12 |
| Company 3 | 16 | 9 | 9 | |
| Company 4 | 18 | | | 13 |
| Company 5 | 16 | 1—stopped | | 13 |

Explanations received from client project managers showed that the first upgrade involved more changes than the second one. Managers accept fulfilling gaps and user requests after the ERP implementation project. CIOs were waiting with business changes, gaps and user requests until the upgrade implementation.

Reimplementation projects in all cases took 12–13 months and managed similarly to the initial implementation projects (see Table 6). Gaps between the first implementation and the reimplementation cause a need to go through process analysis again. Because of the gap of about 10 years, the technical upgrade, in the client's opinion, could not resolve new business demands.

**Table 6.** Average duration (in months) of post-implementation activities in the tested client's environment.

| Scope of Work (SOW) | ERP Extension (on Request) | Patches (Fixes) | Updates | Upgrades | ERP Reimplementation |
|---|---|---|---|---|---|
| Infrastructure installation, company business processes analyses and implementation | | | | | 12–13 |
| Functional changes, optimization in module functionalities with influence on company processes | | | | 3–9 | |
| New functionalities in a tight field, period package with legal changes and bug fixes | | | 1–2 | | |
| Optimization—on request, changes with impact on company business processes or local functionalities | ∞ | | | | |
| Legal changes, bug fixes (often on demand) patches | | 0.25–0.5 | | | |

The right strategic approach can drive sustainability. Companies with an upgrade approach are aware that the outdated ERP system could lead to a decrease in efficiency. CIOs of these companies take these activities seriously and often have procedures to be followed in ERP actualization. On the other hand, there are companies that treat the implementation as a one-off operation. The only activity that was carried out during the maintenance phase was the legal changes installation of the implemented functionalities. However, many years after implementation, the support team had difficulties in maintaining the outdated solution. The biggest challenge was to find specialists who can adapt outdated technology to legal changes.

Data analysis shows that the most time-consuming in the post-implementation perspective are reimplementation projects. Average reimplementation projects are more than

twice as big in the number of modification tasks as upgradess. It is noteworthy that post-implementation activities have less modification tasks in all projects. In all cases, upgrades were also less time-consuming. Based on the the implemented modifications analysis, we can find that reimplementation projects were overestimated to 44% level while upgrades to a 16% level. On the other hand, the underestimation in reimplementations were 31% while in upgrades they were close to 50% of tasks. Differences resulting from the overestimation of modification activities show that organizations are more concerned with reimplementation than with upgrading. The analysis results related to the underestimation of planned and consumed costs confirmed the rule. Regardless of project type, work in this area is undervalued. Interestingly, in terms of service activities such as upgrades, the underestimation is much greater than in the reimplementation project.

Our research report based on reimplementation or upgrade ERP system analyses is rare. Parallel to our research, another article [30] presents a case study of the post-implementation phase related to the ERP system development. Based on their empirical research, authors also noticed that companies reduce the post-implementation phase only to necessary technical activities. In the paper [31], authors present empirical studies in terms of decision support factors in the ERP post-implementation phase. These were the first empirical study in Bangladesh which measured post-ERP implementation for business process performance in organizations. It confirms the essence of this topic and requires future investigation. The main challenge is to reveal reasons of this stage marginalization and to find a solution that will reduce reimplementation projects.

## 5. Discussion

Maintaining business continuity, and sometimes the risk of system downtime, forces companies to constantly develop and adapt the system to business needs. Changes in technology, customer behaviour, and other business environment factors even exact actions in sustainability. ERP post-implementation phase activities are divided into maintenance classes: corrective, adaptive, perfective, preventive, and user support [39]. External Parties help organizations in implementing this maintenance strategy. ERP upgrades are perfectly suited to the perfective maintenance class; on the other hand, reimplementation activities are impossible to be classified in any class directly.

Limiting the budget, avoiding development activities and exclusion of them in obligatory review plans in long-term direct the organization to reimplement the system. It becomes necessary for both technological reasons with the following examples:

- Inaccessible database service;
- Operating system restrictions;
- Environmental limitations (hardware);
- Limitations related to system integration and its expansion necessity.

  And business reasons:

- maladjustment to the current requirements of the business environment;
- Up-to-date optimization;
- support for remote work.

The analysis shows that years of delays, errors, and negligence in the maintenance phase lead to the necessity of reimplementation instead of upgrading the ERP system. This activities are cross-classes because it is at the same time corrective, adaptive and perfective. Data analysis shows that reimplementation should not be understood as reimplementing the same system. All companies accomplished initial implementation successfully and triggered mainly reimplementation actions because of difficulties with adopting the old ERP version with legal and current user demands. Additionally, time cost and task modifications number were 30% lower than in the initial implementation project. The reimplementation approach, compared to the upgrade approach, is around 50% longer and at least twice as large in modification tasks. Despite the higher costs, two companies decided to go through reimplementation. Analysis shows that it occurred in the case of

negligence in the maintenance routine: updates and upgrades. End-user surveys show that, if after initial implementation no upgrades were made after next 2 versions of ERP system, it is required to go through the reimplementation process.

The literature analysis shows that many methodologies end the implementation process at the stage of start-up system. Unfortunately, organizations in many cases have finished the development of the ERP system at this stage and limit future activities only to legal adjustment. The consequence of such actions is the necessity to reimplement the system. In the analyzed literature, these activity scopes are rarely undertaken and do not provide standardized implementation methodologies for reimplementation of well-implemented ERP systems. The paper is based on a case study of five past successfully accomplished ERP implementation projects. It should be highlighted that this kind of IT projects was the most consuming IT activity for these companies both in terms of cost and time consumed, mostly for management staff and senior employees. We collected data through observation, study of formal documentation, and interviews with project managers, system integrators, ERP consultants, CIOs.

In theory, during the maintenance phase, organizations should follow the following activity classes: areas of corrective, adaptive, perfectible, preventive, and user support. In the business practice, the reality looks differently. The interviews conducted among the analyzed companies show that three of them did not follow through with the upgrade project. The main reasons for that state are:

- The initial implementation of the ERP system burdened the organization and subsequent attempts at development and improvement work were reluctantly undertaken;
- In the initial phase of the system maintenance, the client carried out independent developments and carried them out so unskillfully that a system upgrade became impossible;
- The initial implementation contained so many modifications that the system upgrade cost became a serious barrier for the client.

The analysis also shows that business teams have repeatedly ignored the potential opportunities of having the current system version. Actions were usually taken in an unavoidable situation. Results of our analysis could probably be replicated in other countries with similar information technology development as Poland, at least showing the referenced pointers in the paper [40].

## 6. Conclusions

The paper regards problems in the post-implementation phase of management ERP systems. Our research confirms that defining the crucial moment for a technical upgrade is crucial in reducing ERP system maintenance costs. We systematize project activities carried out in the post-implementation stage, both of the correct approach to maintenance and the ERP system upgrade. Our results confirm the essence of introduction the post-implementation methodology for considered industry.

Research results may support subsequent organization decision (CIOs) to upgrade or reimplement the system. The ERP system, implemented without major modifications, enables an efficient upgrade process by up to two system versions (five–six years of application use). The implementation of a comparable system scale with a similar modification number after three released versions of the system requires reimplementation.

Currently, a saturation of the ERP market shows constantly fewer new implementations, unless 81% of companies reach the ERP system. Therefore, the post-implementation phase is so important. It dynamically adjusts the system to the company's needs. In recent years, companies have changed dynamically: they shorten and improve the supply chain, optimize distribution channels, automate settlement methods and communication with government institutions, accelerate production by integrating production machines and devices directly with the ERP system and introduce mass customization manufacturing to reach the most comprehensive possible group of recipients. While the company will maintain the ERP system under the service agreement without a proper methodology and

implementation of long-term plans, emerging needs will not be satisfied. It is necessary to develop a post-implementation methodology. It will significantly reduce upgrade costs and time, thus increasing its availability for other organizations.

The presented research can be extended by examining subsequent reimplementation and post-implementations in companies other than utilities and production. It is possible to consider different original implementation methodologies and develop the research to include companies from other countries. It will provide even more general conclusions and make it possible to propose a universal post-implementation methodology. It will be the subject of our following study.

**Author Contributions:** Conceptualization, A.D.; methodology, A.D., K.G.-D.; validation, K.G.-D., E.K.; formal analysis, A.D.; investigation, A.D.; resources, A.D., E.K.; writing—original draft preparation, A.D., K.G.-D.; writing—review and editing, E.K., J.W.; visualization, K.G.-D.; supervision, J.W.; project administration, E.K.; funding acquisition, J.W. All authors have read and agreed to the published version of the manuscript.

**Funding:** This research is supported by AGH University of Science and Technology subsidy no: 16.16.120.773.

**Institutional Review Board Statement:** Not applicable.

**Informed Consent Statement:** Not applicable.

**Conflicts of Interest:** The authors declare no conflict of interest.

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
