# Peer review of "Post-Implementation ERP Software Development: Upgrade or Reimplementation"

_applsci, doi:10.3390/app11114937_

Round 1

Reviewer 1 Report

I believe the paper is well-written dealing with relevant and important issues.  Nowadays most ERP users are considering upgrade or reimplementation because of vendor EOS (End of Support), cloud-based ERP and, new business requirements.  

I only have some minor suggestions. 

First, regarding Keywords, it seems there are too many. Could you please simplify with a maximum of 5 keywords?  

Second, I think there is some typo. Please correct them accordingly.

Third, it would be great if you could describe theoretical implications and practical implications clearly.   

Thank you very much for the authors' efforts. 

Author Response

Dear Reviewer,

Thank you for the careful reading of the paper and constructive

remarks. We have made the following changes:

  1. In keywords, we decided to omit: "Software product management", "Implementation methodology" and "Critical success factor (CSF)".
  2. Identified typo mistakes are corrected.
  3. We re-build discussion and conclusion paragraphs and add implications caused by maintenance neglect.

Best regards,

The authors

Reviewer 2 Report

This paper provides case studies  based a critical success factor (CSF) categorization. Unfortunately, the authors have identified only trivial minor CSF, type of the post implementation project: upgrade or re-implementation.  This study lacks theoretical and practical value.  The conclusions are already known ones:

 • the ERP system, which is implemented without major modifications, enables an efficient upgrade process by up to two system versions (5-6 years of application use),
 • optimization of the process related to the transfer of modifications between the old and new version of the system could significantly reduce the costs and the actual time of the upgrade.

I would strongly recommend to revisit the CSF and come up with those that can fill the gap in literature and in practice.

Author Response

Dear Reviewer,

Thank you for the careful reading of the paper and constructive
remarks. We have made the following changes:

  1. We agree that this paper shows case studies of 5 projects in the maintenance phase and provides discussion about upgrade/reimplement project – not about CSF of these activities.
  2. We decide to omit keywords with CSF especially because we didn’t consider CSF according to upgrade or reimplementation.
  3. We planned to analyze the effect of lack neglecting the maintenance phase.
  4. We also re-build discussion and conclusion paragraphs and add the theoretical and practical value of this paper.

It is a great suggestion to "revisit the CSF and come up with those that can fill the gap in the literature and in practice". This is an interesting research topic for future research. As we mentioned above our case study provides the analysis of the effects of different approaches to the maintenance phase.

Best regards,

The authors

Reviewer 3 Report

The purpose of this paper is to explore the mechanisms determining the post-implementation approach to upgrade or re-implement the Enterprise Resource Planning (ERP) system. In my opinion, this paper can be accepted for publication after the following revisions have been made:

  1. Suggest rewriting the “Abstract” to briefly describe the motives, purposes, research method, important results and implications, limitations and future research directions of this research.
  2. Suggest enhancing the description of the motive of this research in the “Introduction” section.
  3. Suggest rewriting the “Conclusion” section to give the summary and concluding remarks of this research in more detail.
  4. Suggest citing the related papers published in this journal to link this paper to the material of this journal (no one now).

Author Response

Dear Reviewer,

Thank you for the careful reading of the paper and constructive

remarks. We have made the following changes:

  1. We rewrite the abstract with all suggested issues.
  2. We rewrite the introduction paragraph with the motives.
  3. We re-build discussion and conclusion paragraphs.
  4. We re-build our background and motivation paragraph and add two newest citations to link this paper to the material of this journal.

Best regards,

The authors

Reviewer 4 Report

The purpose of this paper is to examine the methodology and difference understanding to achieve success in the post-implementation stage of ERP systems. The topic of this paper is interesting. However, the paper has to be improved in order to be ready for the publication.

The main strengths of this paper are the following:

  • The title accurately reflects the content of this study.
  • The tables and figures are presented clearly.

First of all, the abstract of the paper is not complete and stand-alone. The authors mentioned the contribution as well as the practical implication of this research. Furthermore, the authors highlighted the need and the research gap in order to conduct this survey and study this research field. However, some details about the methodology and the main findings of the paper should be presented.

The Introduction is not focused. The authors should use the traditional structure, just 4 paragraphs: motivation, gap, method, results, and contributions. The authors did not present the motivation of the paper. It is suggested to improve the flow of this section and highlight the theoretical and practical contribution of the paper. What are the results of previous surveys in this field? The authors mentioned that “The research shows that the lack of software engineering in companies has an impact on their technological dept and enterprises grow. The presented analysis is the basic research to develop a methodology for conducting postimplementation projects based on empirical research. The postimplementation methodology will be used in new industry projects to increase knowledge dissemination and minimize software development gaps. The research clearly shows that the companies which have not upgraded their ERP software for a round 10 years have lost the ability to develop their software and the reimplementation project is obligate”. Unfortunately, the results of this paper do not meet these expectations.

The paper does not demonstrate an adequate understanding of the relevant literature in the field. The authors should analyze the findings and research gaps from previous researchers and not only focus on previous models about ERP implementation. The paper does not present the challenges or the factors that affect each stage of the implementation model.

Section 3 presents the research methodology but it is not clear. This section should be designed based on the existing literature. In terms of methodology, authors should provide more details about the ERP project in each company as the first author was the project manager. What are the main challenges and difficulties during the implementation process? How the post-implementation methodology implemented at each company? It is also strongly recommended to conduct a qualitative survey in order to present the view of each participant in the project.

The findings are a good basis for discussion but the author should answer the following questions: What does this research tell us that we didn’t already know? What new does this paper bring to the table? This paper will have more value if the authors justify the practical and managerial implications of the paper. The conclusion should be supported by the data. Terms such as “Digital transformation” or “SCM” are appeared in this section but they were not analyzed in the manuscript.

There are some language problems in the paper, the text should be checked by an English-speaking person.

Author Response

Dear Reviewer,

Thank you for the careful reading of the paper and constructive remarks.

Remark 1: First of all, the abstract of the paper is not complete and stand-alone. The authors mentioned the contribution as well as the practical implication of this research. Furthermore, the authors highlighted the need and the research gap in order to conduct this survey and study this research field. However, some details about the methodology and the main findings of the paper should be presented.

Replay: We have taken the comments on board to improve and clarify the abstract.

Remark 2: The Introduction is not focused. The authors should use the traditional structure, just 4 paragraphs: motivation, gap, method, results, and contributions. The authors did not present the motivation of the paper. It is suggested to improve the flow of this section and highlight the theoretical and practical contribution of the paper. What are the results of previous surveys in this field? The authors mentioned that “The research shows that the lack of software engineering in companies has an impact on their technological dept and enterprises grow. The presented analysis is the basic research to develop a methodology for conducting postimplementation projects based on empirical research. The postimplementation methodology will be used in new industry projects to increase knowledge dissemination and minimize software development gaps. The research clearly shows that the companies which have not upgraded their ERP software for a round 10 years have lost the ability to develop their software and the reimplementation project is obligate”. Unfortunately, the results of this paper do not meet these expectations.

Replay: We re-write the introduction. We present motivation in a separate paragraph “Background and motivation”. We re-build discussion and conclusions to meet with introduction. Based on 2 projects, the authors showed that organizations attempted to upgrade the system but they failed. Therefore, the thesis can be substituted: "The companies which have not upgraded their ERP software for at least three released next versions of the system, have lost the ability to develop their software and the reimplementation project is obligate". Based on the results, of all projects, we identify the need to develop a new postimplementation maintenance phase methodology, which will be done as part of the next stage of research.

Remark 3: The paper does not demonstrate an adequate understanding of the relevant literature in the field. The authors should analyze the findings and research gaps from previous researchers and not only focus on previous models about ERP implementation. The paper does not present the challenges or the factors that affect each stage of the implementation model.

Replay: We focus only on the post-implementation stage. Importantly, the previous stages affect the activities we analyze in post-implementation projects. We do not consider project management methodology other than for the maintenance phase within ERP systems. ERP systems are very characteristic, therefore most of the top producers introduce their methodologies. It is impossible to easily use methodologies dedicated to other IT projects in these projects.

Remark 4: Section 3 presents the research methodology but it is not clear. This section should be designed based on the existing literature. In terms of methodology, authors should provide more details about the ERP project in each company as the first author was the project manager. What are the main challenges and difficulties during the implementation process? How the post-implementation methodology implemented at each company? It is also strongly recommended to conduct a qualitative survey in order to present the view of each participant in the project.

Reply: All projects were implemented with the usage of the same methodology. Each of the three companies implemented ERP system on the basis of the reimplementation project. They did not use the possibilities offered by the service agreement, e.g. access to updates. Occasionally they used the service agreement only in situations related to legally required changes in the software, e.g. changes in taxes, integration with government institutions. They have lost the ability to develop and upgrade their ERP system. Most of the changes to the system were made independently by the company, often unsuccessful, leading to irreversible changes that made it impossible to perform the upgrade. The paper does not provide additional details about the companies because this is business-sensitive data. Information on the analyzed companies and projects is included in the description and tables.

Remark 5: The findings are a good basis for discussion but the author should answer the following questions: What does this research tell us that we didn’t already know? What new does this paper bring to the table? This paper will have more value if the authors justify the practical and managerial implications of the paper. The conclusion should be supported by the data. Terms such as “Digital transformation” or “SCM” are appeared in this section but they were not analyzed in the manuscript.

Reply: We rebuild the discussion paragraph. We provide a list of our findings according to our projects. We decide to omit the terms “Digital transformation” and “SCM” in the discussion paragraph. These terms were used because of their connection to the implementation project. Our paper describes case studies in the field of post-implementation stage implementation projects in an unprecedented way. One of the goals of this article is to make CIOs aware that gaps and savings in the maintenance stage affect the possible re-implementation costs of the current system.

Best regards,

The authors

Round 2

Reviewer 2 Report

There were no revisions seen in the revised document. The authors' claim there is no post-implementation modelling.  Kindly refer to the following References for latest articles:

A Framework for ERP Post-Implementation Amendments: A Literature Analysis

Benefits Realisation in Post-Implementation Development of ERP Systems: A Case Study
Adapting ERP Systems in the Post-implementation Stage: Dynamic IT Capabilities for ERP

Factors affecting post-implementation success of enterprise resource planning systems: a perspective of business process performance

Author Response

Dear Reviewer,

Thank you for the careful reading of the paper and constructive remarks. 

We apologize for the inconvenience in the previous review. Unfortunately, in the previous version, we misunderstood that the articles should only be from “Applied science”. All current changes have been processed in review mode in blue. 

We are grateful for this literature propositions:

  • A Framework for ERP Post-Implementation Amendments: A Literature Analysis - The paper has already been cited by us in paragraph 2.2.1. ERP Upgrades. Thank you very much for the literature. In edit mode, we made additional changes to the paper.
  • Benefits Realisation in Post-Implementation Development of ERP Systems: A Case Study - In the section “4. Analysis and results” we compare the results of this case study with our research. The authors deal with the analysis in the field of development, while our target was focused on the field of upgrade and reimplementation processes.  
  • Adopting ERP Systems in the Post-implementation Stage: Dynamic IT Capabilities for ERP - As a result of the analysis of the article we rebuild paragraph “2.2 ERP post-implementation activities”. It was indicated which control variables may affect the effectiveness in the post-implementation stage. Importantly, it was shown that the length of the period from implementation to development activities affects the adaptability of the system to business needs. Our article verifies the effectiveness of these activities during the post-implementation stage.
  • Factors affecting the post-implementation success of enterprise resource planning systems: a perspective of business process performance - In the paragraph “4. Analysis and results” we introduce information about research provided in this paper.  

Best Regards

Reviewer 4 Report

Authors have revised some sections of the paper based on the comments. However, major revisions are required in order to improve the readability and presentation of the manuscript.

The main strengths of this paper are the following:

  • The title accurately reflects the content of this study.
  • The tables and figures are presented clearly.

First of all, the abstract of the paper is not complete and stand-alone. The authors added the research gap in order to conduct this survey and study this research field as well as they presented the main findings of the paper. However, the purpose of the paper should be clear. Authors mentioned that “The paper deals with the revealed mechanisms determining the postimplementation approach to upgrade or reimplement the Enterprise Resource Planning (ERP) system. The main objective is to determine the methodology and difference understanding to achieve success in the postimple mentation stage. In particular, we provide a critical success factor (CSF) categorized by the type of the postimplementation project: upgrade or reimplementation”. Please be specific with the objective of the paper. The findings do not present CFS for ERP systems.

The Introduction has been revised. The authors used the traditional structure and improved the flow of this section. However, the authors should add the motivation of the paper in this section and highlight the theoretical and practical contribution of the paper. The authors mentioned that “The paper deals with the revealed mechanisms determining the postimplementation approach to upgrade or reimplement the Enterprise Resource Planning (ERP) system. The main objective is to determine the methodology and difference understanding to achieve success in the postimple mentation stage. In particular, we provide a critical success factor (CSF) categorized by the type of the postimplementation project: upgrade or reimplementation”. Unfortunately, the results of this paper do not meet these expectations and do not present the CFS that affect ERP systems.

Section 2 has considerable been improved but the authors should analyze the findings and research gaps from previous researchers and not only focus on previous models about ERP implementation. The paper does not present the challenges or the CSF that affect each stage of the implementation model.

Section 3 presents the research methodology but it is not clear. This section should be designed based on the existing literature. In terms of methodology, authors should provide references in order to justify the use of this methodology. Authors mentioned that “The data have been collected from interviews with business process managers (BPM), CIO’s, and vendor technical consultants. Data were collected through observation, study of formal documentation, and interviews with project managers, system integrator ERP consultants, CIOs” but they do not present the questions of interviews, the answers of business process managers, CIOs and vendor technical consultant. What is the sample of interviews in each company?

The findings are a good basis for discussion but the authors should answer the following questions: What does this research tell us that we didn’t already know? What new does this paper bring to the table? What are the main challenges and difficulties during the implementation process? How the post-implementation methodology implemented at each company? This paper will have more value if the authors justify the practical and managerial implications of the paper. The conclusion should be supported by the data and provide the limitations of the paper and suggestions for future research.

There are some language problems in the paper, the text should be checked by an English-speaking person.

Author Response

Dear Reviewer,
Thank you for the careful reading of the paper and constructive remarks. We introduce the following changes:  

Remark1: Authors have revised some sections of the paper based on the comments. However, major revisions are required in order to improve the readability and presentation of the manuscript. 

The main strengths of this paper are the following: 

  • The title accurately reflects the content of this study. 
  • The tables and figures are presented clearly. 

First of all, the abstract of the paper is not complete and stand-alone. The authors added the research gap in order to conduct this survey and study this research field as well as they presented the main findings of the paper.  

Answer: Thank you for the review. We rebuild abstract. We hope that the present form of the abstract will meet the expectations. 

Remark 2: However, the purpose of the paper should be clear. Authors mentioned that “The paper deals with the revealed mechanisms determining the postimplementation approach to upgrade or reimplement the Enterprise Resource Planning (ERP) system. The main objective is to determine the methodology and difference understanding to achieve success in the postimple mentation stage. In particular, we provide a critical success factor (CSF) categorized by the type of the postimplementation project: upgrade or reimplementation”. Please be specific with the objective of the paper. The findings do not present CFS for ERP systems. 

 Answer: Thank you for your valuable notice. We removed CFS for ERP systems from the paper because we didn't focus on CSF’s. It is a good idea for a new additional research. 

Instead, we added sentences that better reflect the aim of the paper. All provided changes are in change mode. 

Reark 3: The Introduction has been revised. The authors used the traditional structure and improved the flow of this section. However, the authors should add the motivation of the paper in this section and highlight the theoretical and practical contribution of the paper. The authors mentioned that “The paper deals with the revealed mechanisms determining the postimplementation approach to upgrade or reimplement the Enterprise Resource Planning (ERP) system. The main objective is to determine the methodology and difference understanding to achieve success in the postimple mentation stage. In particular, we provide a critical success factor (CSF) categorized by the type of the postimplementation project: upgrade or reimplementation”. Unfortunately, the results of this paper do not meet these expectations and do not present the CFS that affect ERP systems. 

Answer: Thank you for comments, changes have been made in all mentioned areas. 

Reark 4: Section 2 has considerable been improved but the authors should analyze the findings and research gaps from previous researchers and not only focus on previous models about ERP implementation. The paper does not present the challenges or the CSF that affect each stage of the implementation model. 

Answer:  We appreciate your comments. We made additional verification of the latest literature and supplemented our paper with additional conclusions from the papers: 

  • “Factors affecting post-implementation success of enterprise resource planning systems: a  perspective of business process performance”, Najmul Hasan , May 2019. 
  • “Adapting ERP Systems in the Post-implementation Stage: Dynamic IT Capabilities for ERP”, Neil Chueh-An Lee1, March 2020. 
  • Benefits Realisation in Post-Implementation Development of ERP Systems: A Case Study, Hietala H., 2021. 

CFS analysis for this type of activities will be the subject of further research. 

Remark 5: Section 3 presents the research methodology but it is not clear. This section should be designed based on the existing literature. In terms of methodology, authors should provide references in order to justify the use of this methodology. Authors mentioned that “The data have been collected from interviews with business process managers (BPM), CIO’s, and vendor technical consultants. Data were collected through observation, study of formal documentation, and interviews with project managers, system integrator ERP consultants, CIOs” but they do not present the questions of interviews, the answers of business process managers, CIOs and vendor technical consultant. What is the sample of interviews in each company? 

 Answer: We supplemented the section with “We conducted the research based on five companies using the information collected during the reimplementation or the upgrade projects. Data were collected through observation of steering committees, periodic progress reports, support system data/issue logs, study of formal documentation (meeting reports, protocols, list of change request), and interviews with client project managers, system integrator ERP consultants, CIOs. Due to the diverse scope of the projects, the conducted research could not be carried out based on standardized questionnaire/survey.” 

Remark 6: The findings are a good basis for discussion but the authors should answer the following questions:  What does this research tell us that we didn’t already know?  

Answer: Our research are helpfull the other organizations that are faced with the decision to an upgrade or reimplement their ERP. We answer the question when the organization should do these changes. 

Remark 7: What new does this paper bring to the table?  

Answer: We systematize project activities carried out in the post-implementation stage (in  the scope of maintenance, enhancement and upgrade). 

We identify that in a situation where the above activities are neglected in the long-term, it becomes necessary to reimplement the ERP system. 

 Remark 8: What are the main challenges and difficulties during the implementation process?  

Answer:  Our paper doesn’t deal with the implementation process but post-implementation.  The difficulties and challenges during the implementation process don’t always correlate with the complexity of the post-implementation stage. The implementation projects are well researched while post-implementation isn’t. 

Remark 9: How the post-implementation methodology implemented at each company?  

Answer: The companies kept the ERP system in line base on the service contracts. Without the proper post-implementation methodology or long-term plan. The development of an adequate methodology will be the subject of further research. 

Remark 10: This paper will have more value if the authors justify the practical and managerial implications of the paper.  

Answer: In the research, we answer the question of what are the reasons for the impossibility of upgrading the system in the analyzed organizations. They reflect the limitation of business support by the ERP system.  

We show the risks associated with neglecting the post-implementation stage and impossibility of the further  development of the ERP system. 

 Nowadays, a saturation of the ERP market shows that there will be fewer and fewer new implementations, as 81% of companies already have an ERP system. That is why the post-implementation phase is so important. It dynamically adjusts the system to the company’s needs. In recent years, companies have changed dynamically: they shorten and improve the supply chain, optimize distribution channels, automate the methods of settlements and communication with government institutions, accelerate production by integrating production machines and devices directly with the ERP system and introduce mass customization manufacturing to reach the most comprehensive possible group of recipients. 

Remark 11: The conclusion should be supported by the data and provide the limitations of the paper and suggestions for future research. 

Answer:  We add limitations and planed future research. The limitations are as follows:  

  • a limited number of companies was tested 
  • projects  were carried out on the basis of one methodology 
  • research based on a one system in Poland, in two type of industry: utilities and manufacturing. This system if mainly focus in this areas. 

Best Regards

Round 3

Reviewer 4 Report

Authors have revised some sections of the paper based on the comments. However, major revisions are required in order to improve the readability and presentation of the manuscript.

The main strengths of this paper are the following:

  • The title accurately reflects the content of this study.
  • The tables and figures are presented clearly.

First of all, the abstract of the paper is not complete and stand-alone. The authors removed the research gap and the motivation of this paper. However, the purpose of the paper should be clear. Authors mentioned that “The paper deals with problems in the post-implementation phase of management ERP systems. We present the revealed mechanisms determining the post-implementation approach to upgrade or reimplement the Enterprise Resource Planning (ERP) system. The main aim is to determine the methodology and difference understanding to achieve success in the post-implementation stage. Based on case study analysis, we show that reimplementation means an evolution of the current ERP processes rather than another attempt to "reimplement" an unsuccessful system implementation. The paper deals with the revealed mechanisms determining the postimplementation approach to upgrade or reimplement the Enterprise Resource Planning (ERP) system. The main objective is to determine the methodology and difference understanding to achieve success in the postimple mentation stage. In particular, we provide a critical success factor (CSF) categorized by the type of the postimplementation project: upgrade or reimplementation”. Please be specific with the objective of the paper. Do not repeat the aim of the paper. The findings do not present CFS for ERP systems.

The Introduction has been revised. The authors used the traditional structure and improved the flow of this section. The authors added the motivation of the paper in this section and highlighted the theoretical and practical contribution of the paper.

Section 2 has considerable been improved and the authors analyzed the findings and research gaps from previous researchers. However, it would be interesting to present the challenges or the CSF that affect post-implementation phase.

Section 3 presents the research methodology but it is not clear. This section should be designed based on the existing literature. In terms of methodology, authors should provide references in order to justify the use of this methodology. Authors mentioned that “Data were collected through observation of steering committees, periodic progress reports, support system data/issue logs, study of formal documentation (meeting reports, protocols, list of change request) and interviews with client project managers, system integrator ERP consultants, CIOs” but they do not present the questions of interviews, the answers of business process managers, CIOs and vendor technical consultant. What is the sample of interviews in each company?

The findings are a good basis for discussion but the authors should answer the following questions: What does this research tell us that we didn’t already know? What new does this paper bring to the table? What are the main challenges and difficulties during the post-implementation process? Discussion should not repeat what have been done in the rest of the paper.

There are still some language problems in the paper and the text should be checked by an English-speaking person.

Author Response

Dear reviewer,  

Thank you for your comments. We have introduced the following amendments to the paper: 

Comment 1: First of all, the abstract of the paper is not complete and stand-alone. The authors removed the research gap and the motivation of this paper. However, the purpose of the paper should be clear. Authors mentioned that “The paper deals with problems in the post-implementation phase of management ERP systems. We present the revealed mechanisms determining the post-implementation approach to upgrade or reimplement the Enterprise Resource Planning (ERP) system. The main aim is to determine the methodology and difference understanding to achieve success in the post-implementation stage. Based on case study analysis, we show that reimplementation means an evolution of the current ERP processes rather than another attempt to "reimplement" an unsuccessful system implementation. The paper deals with the revealed mechanisms determining the postimplementation approach to upgrade or reimplement the Enterprise Resource Planning (ERP) system. The main objective is to determine the methodology and difference understanding to achieve success in the postimple mentation stage. In particular, we provide a critical success factor (CSF) categorized by the type of the postimplementation project: upgrade or reimplementation”. Please be specific with the objective of the paper. Do not repeat the aim of the paper. The findings do not present CFS for ERP systems. 

Answer 1:   After the previous review, we have uploaded two versions of the manuscript. First version without marked changes, second version with revisions marked up using the “Track Changes”. In the first version, the following paragraph was not deleted by mistake: 

 “The paper deals with the revealed mechanisms determining the postimplementation approach to upgrade or reimplement the Enterprise Resource Planning (ERP) system. The main objective is to determine the methodology and difference understanding to achieve success in the postimple mentation stage. In particular, we provide a critical success factor (CSF) categorized by the type of the postimplementation project: upgrade or reimplementation” 

We apologize for the mistake. In the paper we do not focus on CSF’s.  

Comment 2: Section 2 has considerable been improved and the authors analyzed the findings and research gaps from previous researchers. However, it would be interesting to present the challenges or the CSF that affect post-implementation phase. 

Answer 2:  We do not consider CFS’s. This is a topic for the continuation of the paper. 

Comment 3: 

Section 3 presents the research methodology but it is not clear. This section should be designed based on the existing literature. In terms of methodology, authors should provide references in order to justify the use of this methodology. Authors mentioned that “Data were collected through observation of steering committees, periodic progress reports, support system data/issue logs, study of formal documentation (meeting reports, protocols, list of change request) and interviews with client project managers, system integrator ERP consultants, CIOs” but they do not present the questions of interviews, the answers of business process managers, CIOs and vendor technical consultant. What is the sample of interviews in each company? 

Answer 3:  All analyzed  post-implementation projects refer to IFS Applications system. The system is dedicated to large companies, especially manufacturing. This ERP system is one of top 10 ERP systems in the world. The producer of this system, like many other producers of the ERP system, provides a methodology for implementation projects, but does not provide a post-implementation methodology.  

The data was collected on the basis of internal document templates. Templates are added to reviewers and editors for viewing as attachments to the manuscript. We cannot publish the entire documentation because this is sensitive information. So, we add to manuscript sentence: “ Information was gathered based on vendor document templates during post-implementations projects managed by first author. “ 

Comment 4: 

The findings are a good basis for discussion but the authors should answer the following questions:  

  1. What does this research tell us that we didn’t already know?  

Answer:  Our results confirm the essence of introduction the post-implementation methodology for considered industry. We add changes to the conclusion. 

2. What new does this paper bring to the table?  

Answer: In the conclusion we have answered the question in sentenses: “Our research confirms that defining the crucial moment for a technical upgrade is crucial in reducing ERP system maintenance costs. We systematize project activities carried out in the post-implementation stage, both of the correct approach to maintenance and the ERP system upgrade “. “The ERP system, implemented without major modifications, enables an efficient upgrade process by up to two system versions (five-six years of application use).” Thus we confirmed that in the considered industry, with a large ERP system, the technological debt is up to 2 releases of a new version of the system.  

3. What are the main challenges and difficulties during the post-implementation process? 

Answer: In the third paragraph of the conclusions we presented challenges and difficulties during the post-implementation process: “It dynamically adjusts the system to company needs. In recent years, companies have changed dynamically: they shorten and improve the supply chain, optimize distribution channels, automate settlement methods and communication with government institutions, accelerate production by integrating production machines and devices directly with the ERP system and introduce mass customization manufacturing to reach the most comprehensive possible group of recipients. While the company will maintain the ERP system under the service agreement without a proper methodology and implementation of long-term plans, emerging needs will not be satisfied.” 

We have introduced linguistic corrections. 

Best regards,

The authors